# Disentangled Safety Adapters Enable Efficient Guardrails and Flexible Inference-Time Alignment

## Abstract

Existing paradigms for ensuring AI safety, such as guardrail models and alignment training, often compromise either inference efficiency or development flexibility. We introduce Disentangled Safety Adapters (DSA), a novel framework addressing these challenges by decoupling safety-specific computations from a task-optimized base model. DSA utilizes lightweight adapters that leverage the base model's internal representations, enabling diverse and flexible safety functionalities with minimal impact on inference cost. Empirically, DSA-based safety guardrails substantially outperform comparably sized standalone models across hate speech classification, detecting unsafe model inputs and responses and hallucination detection with relative improvements of up to 53% in AUC. Furthermore, DSA-based safety alignment allows dynamic, inference-time adjustment of alignment strength and a fine-grained trade-off between instruction following performance and model safety. Importantly, combining the DSA safety guardrail with DSA safety alignment facilitates context-dependent alignment strength, boosting safety on StrongReject by 93% while maintaining 98% performance on MTBench—a total reduction in alignment tax of 8 percentage points compared to standard safety alignment fine-tuning. Overall, DSA presents a promising path towards more modular, efficient, and adaptable AI safety and alignment.

## 1 Introduction

Large Language Models (LLMs) offer immense capabilities Brown et al. (2020); Touvron et al. (2023); Yang et al. (2024), but ensuring their adherence to human-defined safety policies is a critical challenge that is ubiquitous in their development and deployment. Current paradigms for instilling safety in LLMs, primarily safety guardrail models and alignment fine-tuning, present a difficult compromise between development flexibility and inference efficiency.

Safety guardrails are typically deployed as separate classifier models Inan et al. (2023); Han et al. (2024); Ghosh et al. (2024); Zeng et al. (2024); Padhi et al. (2024); Markov et al. (2023); Lees et al. (2022) that analyze the inputs to and/or outputs from an instruction-following LLM, blocking content that violates predefined safety policies. This architectural separation offers significant practical advantages: it allows for independent development and rapid updates of safety mechanisms—crucial when new vulnerabilities are discovered Peng et al. (2024) or when different downstream applications require custom safety policies for the same base LLM. However, the additional inference latency and computational resources required by the separate guardrail model can be prohibitive, especially in streaming and client-side applications or resource-constrained scenarios, often leading to the use of smaller, less performant guardrails.

Conversely, alignment fine-tuning techniques Bai et al. (2022); Thoppilan et al. (2022); Ouyang et al. (2022) integrate safety considerations directly into the LLM's parameters during post-training, improving safety without incurring additional inference costs at deployment. While effective in principle, this approach entangles safety behaviors with the model's general instruction-following abilities. Such entanglement makes targeted safety interventions difficult. For instance, updating the model to address newly identified threats or evolving safety norms would require complete retraining and redeployment of the full instruction-following model, which is highly impractical. Furthermore,

Table 1: Disentangled Safety Adapters (DSA) combine the advantages of existing AI safety methods allowing for independent safety development with low added inference cost.

| Method | Safety Inference Cost | Independent Safety Development |
|---|---|---|
| Separate Guardrails | High | Yes |
| Alignment Training | Low | No |
| DSA (ours) | Low | Yes |

because safety alignment training modifies model behavior for all inputs—irrespective of their safety relevance—it often leads to a degradation in performance on non-safety-related tasks, a phenomenon termed the "alignment tax" Bai et al. (2022); Ouyang et al. (2022), particularly relevant in smaller models.

To address these challenges, we introduce Disentangled Safety Adapters (DSA), a novel framework designed to achieve both flexible safety development and minimal inference overhead. DSA employs lightweight adapter modules that leverage the internal representations of a pre-trained base model to perform safety-specific tasks. Crucially, these adapters operate without altering the base model's parameters or its core inference path for general task computations. This design allows DSA to improve the safety of model response without a significant increase in inference costs, similar to alignment fine-tuning. Simultaneously, like separate guardrail models, DSA factorizes safety-related behavior into distinct, manageable parameters, facilitating independent development and updates (see Table 1).

The DSA framework offers several key advantages: (1) It significantly minimizes inference overhead compared to deploying separate safety models by reusing base model computations, making it ideal for resource-constrained settings such as streaming or client-side applications. (2) It enables flexible, modular development and updating of safety behaviors without requiring retraining or redeployment of the instruction-following base model. (3) It provides a mechanism for dynamic and targeted inference-time adjustment of safety alignment strength, allowing for fine-grained control over the safety-performance trade-off.

Our main contributions are threefold: (i) We demonstrate the efficacy of general disentangled adapters as safety classification guardrails, showing they substantially outperform comparably-sized standalone guardrails and sometimes even match the performance of significantly larger, more costly separate models. We also show they surpass linear probes that are commonly-used for safety classification. (ii) We introduce a method for training disentangled adapters to steer text generation for safety alignment. This enables effective, low-cost, and flexible alignment at the decoding stage, without modifying the base model's learned representations. (iii) We show that combining DSA-based safety classifiers with DSA-based safety alignment facilitates targeted alignment. This approach dynamically adjusts alignment strength based on context, minimizing the alignment tax on general task performance while enhancing safety in high-risk scenarios.

## 2 RELATED WORK

**Disentangled Adapters.** DSA builds upon methods that augment base models while preserving their original parameters. Linear probing Donahue et al. (2014); Alain & Bengio (2016); Belinkov (2022); Buckmann & Hill (2024) provides strong baselines, while more sophisticated architectures like side-tuning Zhang et al. (2020), ladder-side-tuning (LST) Sung et al. (2022), and ResTuning variants Jiang et al. (2023b) demonstrate advantages in flexibility and memory-efficient training. While these methods have proven effective for general tasks, their application to AI safety, specifically for classification and alignment in parallel to instruction-following—remains unexplored. Our work demonstrates their unique suitability for efficient safety guardrails and flexible and targeted alignment.

**Safety Guardrail Models.** Current guardrails (e.g., LlamaGuard Inan et al. (2023), WildGuard Han et al. (2024), AEGIS Ghosh et al. (2024), ShieldGemma Zeng et al. (2024), GraniteGuardian Padhi et al. (2024)) are typically fine-tuned LLMs deployed as separate models, incurring substantial inference costs. Recent work shows promise in leveraging models' internal representations for safety classification via linear probing Sawtell et al. (2024); Marks & Tegmark (2023); Mallen et al.

(2023); Lee et al. (2024); Wang et al. (2024); Stickland et al. (2024), suggesting a path to reduce overhead. Alternatives like LLM Self Defense Phute et al. (2023) or R-tuning Zhang et al. (2024) aim for base model reuse through prompting or fine-tuning but entangle safety with core capabilities. DSA generalizes beyond linear probes to more expressive disentangled adapters, achieving superior classification with minimal inference cost while preserving base model representations.

**Safety Alignment Training.** Safety alignment directly embeds safety behaviors into model parameters Bai et al. (2022); Thoppilan et al. (2022); Ouyang et al. (2022), eliminating inference overhead but creating operational challenges: updating safety requires full model retraining Peng et al. (2024) and induces "alignment tax" on non-safety tasks Bai et al. (2022); Ouyang et al. (2022). Recent decoding-time re-alignment methods Liu et al. (2024); Rafailov et al. (2023); Khanov et al. (2024); Li et al. (2023); Ji et al. (2024) offer flexibility but increase inference cost. Representation engineering approaches Zou et al. (2023); Turner et al. (2023); Zou et al. (2024); Qiu et al. (2024); Bhattacharjee et al. (2024); Arditi et al. (2024); Uppaal et al. (2024); Zhao et al. (2024); Liu et al. (2023); Stickland et al. (2024) can steer generation but directly modify base model activations, requiring separate inference for safety classification when combined with classifiers Stickland et al. (2024). DSA avoids these issues, enabling efficient, flexible alignment without modifying base representations.

# 3 DISENTANGLED SAFETY ADAPTERS (DSA)

The key insight for Disentangled Safety Adapters (DSA) is that safety classification and alignment always need to be addressed in parallel to any other generative task.

Consider a base model $B(\mathbf{x}|\boldsymbol{\theta}) = \hat{\mathbf{y}}$ with parameters $\boldsymbol{\theta}$ that is trained to generate outputs $\mathbf{y}$ from inputs $\mathbf{x}$ to solve a specific task. Ensuring the safety of outputs $\hat{\mathbf{y}}$ can be achieved through a separate guardrail model, alignment training or a combination of both.

**DSA Safety Guardrail.** In general, a separate safety guardrail model $S(\mathbf{x}, \mathbf{y}|\boldsymbol{\phi}) = \hat{s}$ with parameters $\boldsymbol{\phi}$ is trained to classify model inputs and outputs as safe ($s = 1$) or unsafe ($s = 0$) (Fig. 1a). This approach isolates safety functionality in parameters $\boldsymbol{\phi}$ from the base model's task-specific parameters $\boldsymbol{\theta}$. The base model's original performance is preserved for data deemed safe. However, the added computation of the safety classifier increases inference costs.

Disentangled Safety Adapters (DSAs) also isolate safety-specific computations within independent parameters $\boldsymbol{\phi}$ while maximizing the reuse of base model computations (Fig. 1b). A DSA safety classifier is defined as:

$$S_{\text{DSA}}(\mathbf{h_b}(\mathbf{x}, \hat{\mathbf{y}}|\boldsymbol{\theta})|\boldsymbol{\phi}) = \hat{s} . \tag{1}$$

This leverages internal representations, $\mathbf{h_b}(\mathbf{x}, \hat{\mathbf{y}}|\boldsymbol{\theta})$, from the input and response as additional inputs to achieve comparable safety classification performance to standalone guardrail models the size of the base model. At the same time it dramatically reduces computational overhead and inference cost, as demonstrated in Section 4.1.

**DSA Safety Alignment.** Standard safety alignment training modifies the base model's parameters to directly produce safe outputs: $B(\mathbf{x}|\boldsymbol{\theta}_{\text{safe}}) = \hat{\mathbf{y}}_{\text{safe}}$ (Fig. 1c). This avoids increased inference costs, as no additional computation is introduced. However, safety and non-safety behavior are entangled in the same set of parameters, making it challenging to independently update and adapt safety behavior when previously unknown issues are detected. Additionally it can impact model behavior even on non-safety-related data, often resulting in a decrease in task performance termed alignment-tax Bai et al. (2022); Ouyang et al. (2022).

DSA safety alignment optimizes the parameters $\boldsymbol{\phi}$ of a disentangled adapter to predict adjustments to the logits of a base model such that the output of the resulting model $M$ is safe:

$$M(\mathbf{x}, \lambda|\boldsymbol{\phi}, \boldsymbol{\theta}) = \text{softmax}\left[\lambda \mathbf{z}_b(\mathbf{x}|\boldsymbol{\theta}) + (1 - \lambda)\mathbf{z}_{\text{DSA}}(\mathbf{h_b}(\mathbf{x}|\boldsymbol{\theta})|\boldsymbol{\phi})\right] = \hat{\mathbf{y}}_{\text{safe}}, \tag{2}$$

where $\mathbf{z}_b$ are the base model's logits, $\mathbf{z}_{\text{DSA}}$ are the logits predicted by the DSA and $\lambda$ is an interpolation parameter, bound between 0 and 1, that determines the degree with which the base model's logits are reused during training. This can effectively align the resulting model's outputs with negligible

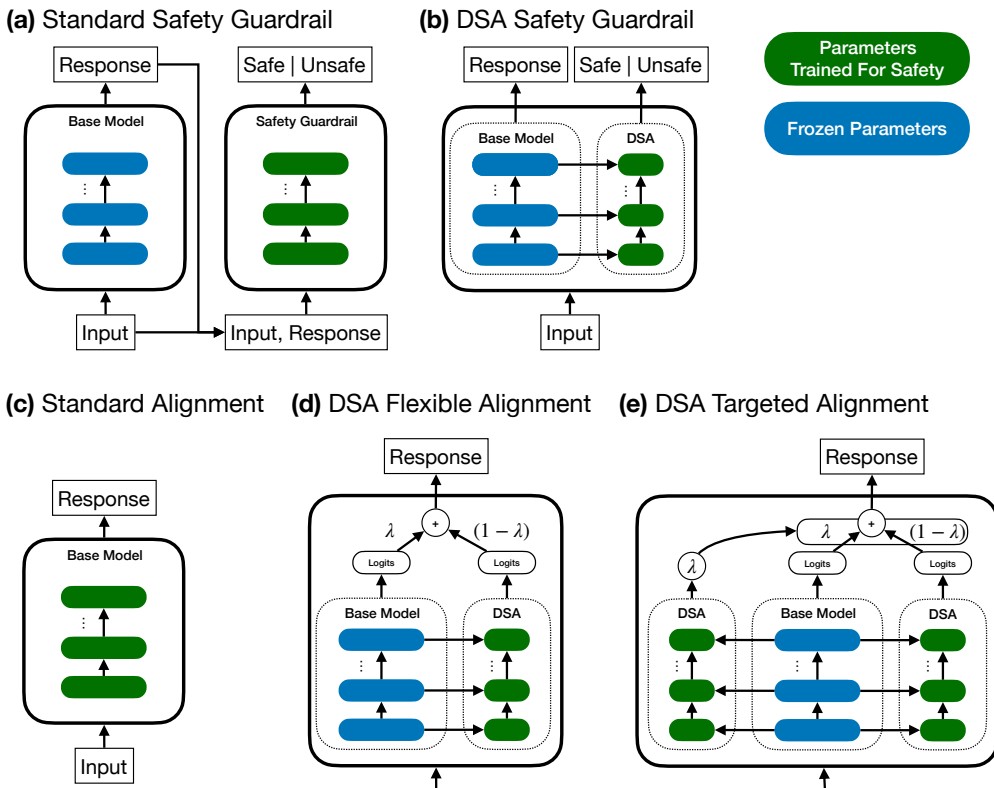

Figure 1: Overview of DSA architecture and how it compares to standard safety techniques. **(a)** Standard safety guardrails are separate models trained to classify model inputs and responses as either safe or unsafe without re-using the computations of the generator base model. **(b)** DSA safety guardrails reuse the computations of the generator base model to improve performance with low added inference costs. **(c)** Standard safety alignment training fine-tunes the base model's parameters with safety data. **(d)** DSA alignment training optimizes a separate set of parameters while re-using the base model's computations, which enables flexible interpolation between base model behavior and safety-aligned behavior. **(e)** Combining the DSA safety classifier and the DSA alignment training, allows for context-dependent adjustment of alignment strength, to minimize the impact of safety alignment on model performance for non-safety related data.

additional computation as shown in Section 4.2. At the same time it provides the advantage of fully disentangling the safety-specific from the task-specific computations, which enables independent updating of safety behavior as well as inference-time flexibility of alignment strength. In particular, Liu et al. (2024) described how to flexibly tune the strength of safety alignment at inference time by linearly interpolating between the logits of a unaligned and those of an aligned model. DSAs facilitate this flexible alignment by simply varying the $\lambda$ parameter at inference time without the doubled inference cost or the need of any model retraining.

**DSA Targeted Alignment.** Finally, decoupling safety alignment from the base model into the DSA allows for context-dependent adjustment of alignment strength. We can use a DSA safety classifier with parameters $\phi_c$ to detect whether the input $\mathbf{x}$ to the model is unsafe and thus determine the interpolation weight $\lambda$ between the not-safety-aligned base model with parameters $\theta$ and the safety-alignment DSA with parameters $\phi_a$ (Fig. 1e). The interpolation weight $\lambda$ can be any function $f$ of the output of the safety classifier:

$$\lambda(\mathbf{x}|\phi_c) = f\big(S_{\text{DSA}}(\mathbf{h_b}(\mathbf{x}|\theta)|\phi_c)\big) = f\big(\hat{s}|\phi_c\big), \tag{3}$$

but in Section 4.2 we focus on using a step function to switch-on the DSA safety alignment whenever the input to the model is classified as unsafe.

The full model $M_{\text{TA}}$ with targeted safety alignment is given by.

$$M_{\text{TA}}(\mathbf{x}|\boldsymbol{\phi_c}, \boldsymbol{\phi_a}, \boldsymbol{\theta}) = \text{softmax}\left[\lambda(\mathbf{x}|\boldsymbol{\phi_c})\mathbf{z}_b(\mathbf{x}|\boldsymbol{\theta}) + (1 - \lambda(\mathbf{x}|\boldsymbol{\phi_c}))\mathbf{z}_{\text{DSA}}(\mathbf{h_b}(\mathbf{x}|\boldsymbol{\theta})|\boldsymbol{\phi_a})\right]. \quad (4)$$

This ensures that safety alignment impacts only model responses if the context is safety-relevant, mitigating the alignment tax on model performance (Section 4.2).

## 4 EXPERIMENTS AND RESULTS

### 4.1 GUARDRAIL MODELS

**Datasets.** We evaluate DSA safety guardrails on several safety classification tasks. We use the ToxiGen dataset to evaluate the ability to detect subtle and implicit hate speech Hartvigsen et al. (2022). For evaluating the ability to detect unsafe user queries, we use the prompt labels from AEGIS2.0 dataset Ghosh et al. (2025). To evaluate the ability for detecting unsafe model responses, we use the BeaverTails-30k dataset Ji et al. (2023). Since preventing hallucinated content is a critical component of safety guardrails, we use the Summedits dataset Laban et al. (2023) to test the ability to detect hallucinated content in model responses. Full details on datasets can be found in Appendix D.

**Models.** To evaluate DSA safety guardrails across multiple model families and scales, we use a variety of instruction-tuned models as the base model. We use Qwen2.5 7B and 14B models Yang et al. (2024), Mistral-7B-v0.3 (Jiang et al., 2023a) and Gemma2-9B (Team et al., 2024).

We compare against two different separate guardrail models: a model with high added inference cost that matches the size of the base model and a model with low added inference cost that matches the size of the disentangled adapter. For the high-cost model we use LoRA adaptation Hu et al. (2022) to fine-tune the base model for each safety classification task. Since LoRA fine-tuning changes all intermediate activations of the base model, no computations from the base model's response can be reused at inference time and the whole 7B parameter model needs to be re-run to make the safety classification on the base model's output. For the low cost model, we reduce the size of the base models' hidden and intermediate states by a factor of 16 and the number of attention and key-value heads by a factor of 2. The low cost model is initialized by distilling from the base model using the Dolma and Tulu-3 datasets Soldaini et al. (2024); Lambert et al. (2024) and then fully finetuned on the safety classification tasks.

The defining feature of a DSA guardrail is that it runs in parallel to the base model, while leveraging base model's internal representations for the safety classification task. To compare to the separate guardrail models, we evaluate multiple disentangled adapter architectures with increasing complexity on the given safety classification tasks.

First, as a baseline, we train an L2-regularized logistic regression classifier from the base model's hidden representations. This is the simplest version of a DSA and its efficacy for safety classification tasks is widely acknowledged (e.g. Sawtell et al. (2024)) (Table 2, DSA:PLR).

Second, we increase the complexity of the DSA by using a stack of bottleneck feed-forward layers with a non-linearity Houlsby et al. (2019), which run in parallel to the base model and consume the base model's representations at every layer. This follows the Res-Tuning-Bypass adapter architecture, that was previously applied to non-safety-related image classification and generation tasks Jiang et al. (2023b) (Table 2, DSA:RTB).

Finally, as the most complex disentangled adapter architecture in the comparison, we follow the ladder side tuning architecture that was previously applied to non-safety related NLP and vision-language tasks Sung et al. (2022). In ladder side tuning, a down-sized version of the base model is run in parallel to the base model, with additional connections to consume the base model's representations at every layer. We use the down-sized base model as the parallel side-network that we initialize by distilling from the large base model (Table 2, DSA:LST).

Full details of all training and cross validation hyperparameters can be found in Appendix B.1.

**Results**

Overall, we find that Disentangled Safety Adapters (DSAs) demonstrate substantial improvements over standalone guardrail models that operate with comparable low inference costs (Table 2). For

Table 2: Results for Guardrail Models. Test AUC comparison of different safety classifiers across various datasets. Each method use less that 1% extra FLOPs over the base model except the large separate guardrail which uses 100% extra FLOPs (see Appendix C for detailed FLOP calculations).

| Base model | Method | ToxiGen | AEGIS2.0 | BeaverTails | Summedits |
|---|---|---|---|---|---|
| Qwen2.5-7B | Separate Guardrail 25M | 0.92 | 0.90 | 0.90 | 0.61 |
| | DSA:PLR Sawtell et al. (2024) | 0.97 | 0.90 | 0.90 | 0.83 |
| | DSA:RTB Jiang et al. (2023b) | **0.98** | 0.92 | 0.91 | 0.83 |
| | DSA:LST Sung et al. (2022) | **0.98** | **0.93** | **0.93** | **0.88** |
| | Separate Guardrail 7B | 0.96 | 0.90 | 0.91 | 0.88 |
| Qwen2.5-14B | Separate Guardrail 51M | 0.92 | 0.90 | 0.89 | 0.62 |
| | DSA:PLR | 0.98 | 0.92 | 0.91 | 0.81 |
| | DSA:RTB | 0.98 | 0.91 | 0.90 | 0.83 |
| | DSA:LST | **0.99** | **0.94** | **0.92** | **0.88** |
| | Separate Guardrail 14B | 0.96 | 0.90 | 0.91 | 0.88 |
| Mistral-7B-v0.3 | Separate Guardrail 27M | 0.92 | 0.90 | 0.90 | 0.59 |
| | DSA:PLR | 0.96 | 0.89 | 0.85 | 0.62 |
| | DSA:RTB | **0.98** | 0.92 | 0.91 | 0.72 |
| | DSA:LST | **0.98** | **0.93** | **0.93** | **0.90** |
| | Separate Guardrail 7B | 0.99 | 0.94 | 0.94 | 0.92 |
| Gemma2-9B | Separate Guardrail 31M | 0.92 | 0.89 | 0.89 | 0.59 |
| | DSA:PLR | 0.97 | 0.88 | 0.89 | 0.70 |
| | DSA:RTB | 0.96 | 0.91 | 0.91 | 0.83 |
| | DSA:LST | **0.98** | **0.93** | **0.93** | **0.89** |
| | Separate Guardrail 9B | 0.99 | 0.94 | 0.94 | 0.93 |

hate speech classification on ToxiGen the best DSA architectures achieve 0.99 AUC and clearly outperform the 0.92 AUC of the low-cost separate guardrail. In the task of detecting unsafe model inputs and outputs, the DSA:LST model excels, yielding between 0.92 and 0.93 AUC on both the AEGIS2.0 and BeaverTails datasets compared to the 0.89 and 0.90 AUC achieved by the low-cost separate guardrail. The most pronounced advantage of DSA is observed in hallucination detection on the Summedits dataset. Here, the DSA:LST models achieve between 0.88 and 0.90 AUC, a dramatic increase from the low-cost guardrail's 0.59 to 0.62 AUC. This improvement of up to 53% in AUC underscores DSA's effectiveness in re-using the powerful representations from the base model, particularly for more complex safety reasoning tasks like identifying hallucinations.

Furthermore, even when comparing against the high-cost separate guardrails, the best performing DSA models (LST) achieve comparable results. On Summedits, our most complex task, DSA:LST matches performance of the much more computationally expensive high-cost guardrail for Qwen models (0.88 AUC) and performs only slightly worse for Mistral (0.90 vs. 0.92 AUC) and Gemma (0.89 vs. 0.93 AUC) models.

The results also indicate a clear trend: increasing the sophistication of the DSA architecture, from the simpler PLR to RTB and then to LST, generally yields progressive improvements in AUC across all evaluated safety tasks. This highlights the utility of the DSA framework, which generalizes the commonly employed linear probing of the base model's representations for safety classification.

In summary, DSAs provide highly effective safety guardrails, achieving performance in the range of costly, full-sized guardrail models, but at almost no additional computational cost ($< 1\%$ additional FLOPs, see Appendix C for detailed FLOP calculations).

## 4.2 SAFETY ALIGNMENT

**Datasets.** In this section we train models for safety alignment using the helpfulness and harmlessness human preference data from the Anthropic hh-rlhf dataset Bai et al. (2022). We evaluate the impact of safety alignment training on a model's conversational and instruction-following ability using MT-Bench Zheng et al. (2023) and MMLU (Hendrycks et al., 2021). To assess the impact of safety

alignment training on a model's safety behavior, we use StrongREJECT Souly et al. (2024) and JailbreakBench (Chao et al., 2024). Full details for all datasets can be found in Appendix D.

**Models.** To cleanly isolate and measure the potential of the DSA framework for safety alignment training, it is critical to use a base model that has not already undergone safety fine-tuning. Therefore, we use Zephyr-7B Tunstall et al. (2023) as the base model, which is optimized for human intent alignment but lacks explicit safety training, making it an ideal candidate for studying safety alignment interventions.

As a standard safety alignment baseline, we directly fine-tune Zephyr-7B using LoRA Hu et al. (2022). This method (Fig. 2, "LORA") incurs no additional inference cost but entangles safety with base model computations, precluding flexible alignment strength adjustment without running both aligned and unaligned models Liu et al. (2024).

Our initial DSA alignment model employs the DSA:LST architecture Sung et al. (2022), which excelled in our safety classification experiments (Section 4.1). The side network uses the same down-scaled Qwen2.5 7B architecture but based on Zephyr's parameter dimensions and layer count for connections at each base model layer. The LST architecture directly predicts the aligned model response ($\lambda = 0$ during training) and thus needs to be initialized for high quality text generation. Therefore, similar to the DSA:LST safety classifier, we initialize the side network via distillation from the base model but now we also include the connections to the base model as trainable parameters during distillation (Fig. 2, "DSA:LST").

Finding the original LST's generation quality limited for alignment, we introduced modifications to enhance it while maintaining a small adapter size. Firstly, inspired by Side-Tuning Zhang et al. (2020), we changed the adapter to predict a residual to the base model's logits (set $\lambda = 0.5$ during training) rather than generating responses directly. Consequently, instead of distillation, this "DSA:LST+" adapter is initialized via standard next-token prediction on the Dolma Soldaini et al. (2024) and Tulu-3 Lambert et al. (2024) datasets. Secondly, we optimized how the DSA consumes base model representations, finding that low-rank projections followed by cross-attention yielded the best results (Fig. 2, "DSA:LST+").

The DSA:LST and DSA:LST+ architectures cannot perfectly initialize from base model generations for alignment and their generative capacity might be limited by the dimensionality of their disentangled representations. To address these limitations, we introduce DSA:Last-Layers-Duplicate (DSA:LLD). This architecture duplicates the final $N$ layers of the base model, fine-tuning only these duplicated layers for safety alignment using LoRA. By design, DSA:LLD initializes with base model behavior, reuses all computations up to the duplicated segment, and only adds overhead for these final layers. While slightly more expensive than previous DSAs (each duplicated layer adds 3% computation versus <1%), we found duplicating $N = 2$ layers offered a favorable performance-cost trade-off (Fig. 2, "DSA:LLD").

Finally, we implement targeted alignment (Section 3, Eq. 4) for both the DSA:LST+ and DSA:LLD alignment adapters by combining them with the DSA:LST safety classifier from Section 4.1. This classifier, trained on AEGIS2.0 with Zephyr-7B as its base, categorizes model inputs as safe or unsafe. For inputs classified as safe, the interpolation parameter $\lambda$ is set to 1 (base model only). For unsafe inputs, $\lambda$ is set to the respective interpolation weight, blending base model and DSA adapter logits (Fig. 2, "DSA:LST+:TA" and "DSA:LLD:TA"). Importantly, this entire process—classification and conditional alignment—occurs efficiently within a single base model inference pass, as both DSA modules reuse base model representations with minimal added computation.

All alignment models, including our DSA variants, were fine-tuned using the DPO objective Rafailov et al. (2023) on the Anthropic hh-rlhf dataset Bai et al. (2022). Full training details and hyperparameters are provided in Appendix B.2.

**Results**

Standard LoRA fine-tuning for alignment establishes a strong baseline, reducing the StrongReject score by 93% while retaining 90% of the base model's MTBench performance (Fig. 2, "LORA", $\lambda = 0$). Consistent with Liu et al. (2024), linearly interpolating logits (varying $\lambda$) allows a smooth trade-off between the base and LoRA-aligned model, albeit at twice the inference cost.

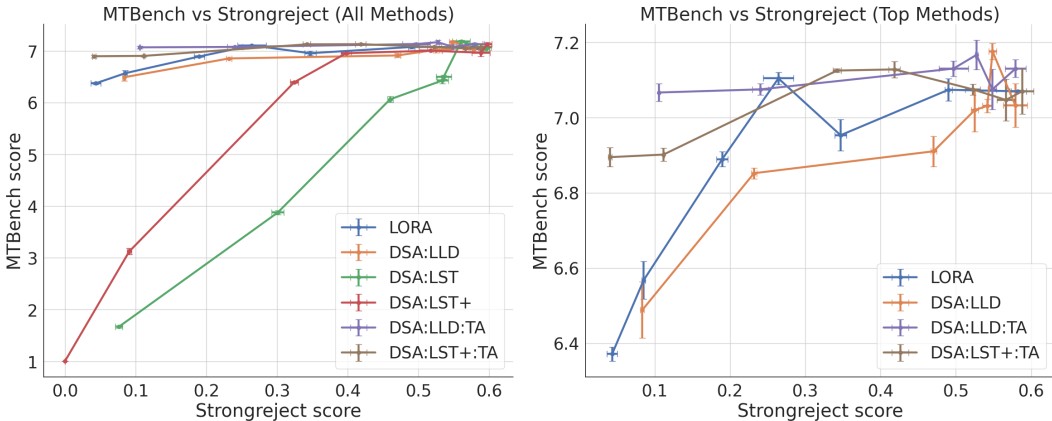

Figure 2: DSA Flexible Alignment Results. Scores achieved by different adapters on MTBench (y-axis, higher is better) and StrongReject benchmark (x-axis, lower is safer). Different degrees of alignment for each method are achieved by varying interpolating strength $\lambda$ between logits from the base model and adapter (DSAs) or adapted model (LORA). The leftmost point for each method corresponds to $\lambda = 0$ with no direct contribution of the base model to the logits. Targeted alignment (TA) using two DSAs simultaneously achieves comparable safety levels to the LoRA alignment training baseline while significantly reducing the alignment tax on MTBench performance. Results are averaged over 3 runs of the llm-as-a-judge evaluation for both StrongReject and MTBench and error bars indicate the 95% CI for the standard error of the mean.

Table 3: Targeted Alignment Results. Comparison of DSA-based targeted safety alignment at $\lambda = 0$ with LoRA-based alignment on safety and capability benchmarks. MMLU and MTBench measure general model performance on a wide domain of tasks. StrongReject and JailbreakBench measure the safety of model responses on a set of harm-eliciting prompts. DSA:LLD:TA preserves nearly all base model capabilities while dramatically increasing safety performance. DSA:LST+:TA matches or exceeds the safety performance of the LoRA-aligned Zephyr model while preserving substantially more of the base model's general capabilities.

| Model | MTBench ↑ | MMLU ↑ | StrongReject ↓ | JailbreakBench ↑ |
|---|---|---|---|---|
| Zephyr Base Model ($\lambda = 1$) | **7.07** | **51.20** | 0.58 | 15 |
| Zephyr LoRA-aligned ($\lambda = 0$) | 6.37 | 45.99 | **0.04** | 66 |
| DSA:LLD:TA ($\lambda = 0$) | **7.07** | 50.15 | 0.11 | 55 |
| DSA:LST+:TA ($\lambda = 0$) | 6.89 | 48.69 | **0.04** | **74** |

The initial DSA:LST model struggled with instruction-following quality, shown by poor MTBench scores, which also contributed to a low StrongReject score (as StrongReject scores low on unhelpful responses to unsafe prompts) (Fig. 2, "DSA:LST"). The improved DSA:LST+ model performed considerably better, nearly matching the LoRA model at medium $\lambda$ values (trained with $\lambda = 0.5$ to predict logit adjustments). However, even at its optimal $\lambda = 0.4$, DSA:LST+ only reduced the StrongReject score by 45% while maintaining 91% MTBench performance; further decreasing $\lambda$ quickly degraded MTBench scores (Fig. 2, "DSA:LST+"). Both DSA:LST variants only add <1% inference overhead, and enable $\lambda$-interpolation at no extra inference costs.

The DSA:LLD architecture, designed to address the limitations of DSA:LST/LST+, performed comparably to LoRA fine-tuning. It reduced StrongReject vulnerability by 86% while maintaining 92% MTBench performance (Fig. 2, "DSA:LLD", $\lambda = 0$). Duplicating the final two base model layers, DSA:LLD incurs about 6% additional inference cost over the base model. However, unlike LoRA, it also allows flexible $\lambda$-interpolation for dynamic alignment strength at no extra computational cost beyond the initial adapter overhead.

Targeted alignment demonstrates the full potential of the DSA framework and shows very favorable performance against the LoRA baseline, especially at full alignment strength with $\lambda = 0$ (Table 3).

Combining the DSA:LST+ alignment adapter with the AEGIS2.0-trained DSA:LST safety classifier reduced StrongReject vulnerability by 93% and increased JailbreakBench safety from 15% to 74% while preserving 98% of MTBench and 95% of MMLU performance (Table 3, "DSA:LST+:TA". This matches or exceeds LoRA's safety improvement but reduces its alignment tax by 5 to 8 percentage points, all with less than 2% added inference compute. Furthermore, targeted alignment with the DSA:LLD model (DSA:LLD:TA) improved StrongReject safety by 83% and increased Jailbreak-Bench safety from 15% to 55% while preserving 98% of MMLU and full MTBench performance (Table 3, "DSA:LLD:TA"), using less than 8% added inference cost.

Overall, these results demonstrate that the DSA framework, particularly through targeted alignment, can achieve safety levels comparable to traditional alignment fine-tuning while significantly mitigating the alignment tax and maintaining remarkable inference efficiency.

## 5 LIMITATIONS AND FUTURE WORK

While Disentangled Safety Adapters (DSA) offer a promising approach for efficient and modular AI safety, our current work has some potential limitations that also point towards exciting avenues for future research.

First, while we see that our more complex DSA Guardrails considerably outperform widely-used linear probes, we do acknowledge the engineering overhead and design considerations involved in implementing these more complex adapters.

Second, our exploration was limited to combining at most two DSA modules: a general safety classifier and an alignment module. Future research could investigate the composition of multiple, diverse adapters. This includes, for instance, combining input classifiers for jailbreak attempts with output classifiers for various harms (e.g., unsafe responses, hallucinations), or integrating multiple alignment adapters for nuanced steering or task-specific fine-tuning. Developing strategies for the effective and efficient composition of such multi-adapter systems is a key future direction.

Third, in alignment training, our smallest disentangled adapters (DSA:LST, DSA:LST+) fell short of matching the performance of direct base model fine-tuning (Fig. 2). While the DSA:LLD approach offered substantial improvement, further research into optimal disentangled adapter architectures specifically for performant language generation is an important future direction.

Finally, the efficacy of our proposed targeted alignment relies on the DSA safety classifier's accuracy in identifying unsafe contexts. A key advantage of DSA's modularity is that if the classifier errs, it can be improved or replaced in a targeted manner. Nevertheless, future work could explore to use a smooth function of the logit of the classifier as the interpolation weight and thus tune the alignment strength dynamically proportional to the probability of the input being unsafe. This would increase the robustness of the targeted intervention albeit at the cost of a less factorized system.

## 6 CONCLUSION

In this work, we introduced Disentangled Safety Adapters (DSA), a novel framework uniquely designed to integrate safety functionality efficiently alongside an LLM's primary tasks. A key practical achievement of DSA is the ability to deploy safety guardrails at the performance level of the fine-tuned base model at a fraction of the inference cost. Beyond guardrailing, DSA's modular architecture enables transparent steering of model outputs for safety alignment and flexible, inference-time adjustment of alignment strength, without requiring costly retraining of the base model. We further demonstrated that combining classification and alignment DSAs facilitates context-dependent alignment to significantly reduce the common alignment tax on general task performance. Overall, DSA offers a more modular, efficient, and adaptable paradigm for building safe and aligned AI systems.

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

**LLM Usage in Writing**    The authors used LLMs during the editing process of this manuscript to improve clarity of wording and revise potential grammatical mistakes.

## A    BROADER IMPACTS

Disentangled Safety Adapters (DSA) are introduced to enhance AI safety by providing efficient and flexible mechanisms for guardrails and safety alignment. The computational efficiency of DSA-based safety guardrails can enable more robust safeguards, particularly in resource-constrained environments where deploying costly separate models is prohibitive. DSA's factorized design, separating safety logic from the base model, empowers developers to rapidly respond to new safety challenges and to mitigate harmful outputs with potentially less degradation of the model's general

utility (i.e., a reduced alignment tax). However, a critical consideration for DSA-based targeted alignment is the quality of the safety classifier; an inadequately developed or poorly tested classifier could inadvertently misguide the alignment process, potentially even reducing overall safety or introducing new biases. Furthermore, like all safety mitigations, DSA-based mechanisms are not infallible and will likely possess limitations and vulnerabilities that could be exploited or lead to failures in novel situations. Responsible development and deployment therefore requires continuous monitoring of model behavior and proactive investigation to identify vulnerabilities, both pre- and post-deployment. While DSA offers a promising advancement for modular and efficient AI safety, its application necessitates careful ethical consideration, robust policy definition, and its integration as one component within a multi-layered approach to responsibly building safe AI systems.

## B  FULL EXPERIMENT DETAILS

### B.1  SAFETY GUARDRAIL EXPERIMENTS

For each of the 4 datasets used in Guardrail experiments, we train models for 20 epochs with early-stopping based on loss on held-out validation set, with a patience of 3 epochs. When using LoRA adapters on the base model, we set the rank of the adapters to 32.

The side-network of the LST adapters are initialized using distillation with the base model as the teacher. The side-network uses a scaled down version of the base model with hidden and intermediate state dimensions scaled down by $16\times$ and the number of attention heads and key-value heads are scaled down by $2\times$. For better initialization, we share the input embedding lookup table and the output language model head of the adapter with the base model (as outlined in the original ladderside tuning work). Since the architectures of different base models (e.g. Qwen, Mistral etc.) are largely similar transformer stacks, for simplicity of implementation, we instantiate the side network in LST and LST+ adapters using the architecture of the Qwen model class.

For all the distillation training runs, we first distill using the Dolma corpus for 5000 steps where each step has an effective batch size of 1024 sequences. The maximum sequence length of each training example is 1024. We use a cosine learning schedule with peak learning rate of 6e-4 and with 10% warmup steps. We further distill the resulting model on 500K datapoints from the Tulu-v3 SFT mixture, with batch size of 256, maximum sequence length of 2048, and a linear learning rate schedule with peak learning rate of 2e-6 and warmup for the initial 3% steps.

We observed high variance in training stability when using DSAs on Mistral and Gemma models, and so we carried out 3 training runs with different seeds for each experiment involving them (including LoRA fine-tuning) and evaluated the best checkpoint selected based on validation set performance.

The batch size is kept as 32 for all training runs. All training is carried out in 16-bit precision using the 'bfloat16' datatype for parameters. We use Adam optimizer and tune the learning rates among the values of 1e-5, 5e-5 and 1e-4 for all methods except logistic regression. For logistic regression, we use SGD optimizer with L2 regularization coefficient of 0.01. For logistic regression, we also include a higher learning rate of 0.001 in the hyperparameter search and we also tune for the best layer to get features for logistic regression. For Qwen-2.5-7B-Instruct model we tune using layers 4,8,12,16,20,24,28. For Qwen-2.5-14B-Instruct we use layers 8,16,24,32,40,48. For Gemma2-9B-it model we use layers 11,21,32,42. For Mistral-7B-Instruct-v0.3 we use layers 8,16,24,32.

When training Separate Guardrails either using standard LORA adapters on top of the base model or finetuning a separate scaled-down model, we prepend a task-specific instruction to the input to be graded and the model's target is ground truth answer is either a 'Yes' or a 'No'. We use the probability of generating 'Yes' in the first timestep of the model's response, divided by the sum of probabilities of 'Yes' and 'No' tokens, as the predicted probability of harm. The instructions used for each dataset are given in Table 4. However, for the disentangled adapter architectures, we do not use any instructions as given in Table 4 since that would amount to interfering in the base model's computations by injecting a safety-specific prompt. Hence, in experiments with disentangled adapters, we feed the user query and optionally model response (in the datasets where it exists such as BeaverTails) into the model as-is without modification, only applying the standard chat template.

For Res-Tuning-Bypass architecture, we use a bottleneck hidden dimension of 32 for the adapter layers used in it.

Table 4: Prompts used with Separate LORA-adapted Guardrails on different safety classification datasets

| Dataset | Instruction |
|---------|-------------|
| AEGIS2.0 | You are given a prompt from a human. Your task is to detect if the prompt is harmful. Prompt: [PROMPT] Is the given prompt harmful? |
| ToxiGen | You are given a statement from a human. Your task is to predict if the statement is toxic. Statement: [STATEMENT] Is the statement toxic? |
| BeaverTails | You are given a prompt from a human and a response from a language model below. Your task is to detect if there is any harmful content in the language model's response. Prompt: [PROMPT] Response: [RESPONSE] Does the language model's response contain harmful content? |
| Summedits | You are given a document and its summary below. Your task is to predict if there is a factual inconsistency in the summary or not with respect to the reference. Document: [DOCUMENT] Summary: [SUMMARY] Does the summary have any factual inconsistency or hallucination with respect to the document? |

## B.2 SAFETY ALIGNMENT EXPERIMENTS

In all DPO training runs, the reference model is set to the be the base model with the initialized (and frozen) adapter, and the model under training is a clone of the same model and adapter with only the adapter weights being trainable and we set the number of training epochs to 10 and batch size to 32. We tuned for learning rate by trying out peak learning rates of 5e-5, 1e-4 and 5e-4 with a cosine learning rate schedule and a warmup phase of first 10% of steps. We found that generally higher learning rates tended to perform better. We used a value of 0.1 for the beta parameter in the DPO trainer that controls the relative weight of kl-divergence with-respect-to the unaligned model's outputs in the loss term. For alignment using standard LORA adapters, we try out ranks 8, 16, 24 and 32. All training is carried out in 16-bit precision using the 'bfloat16' datatype for parameters.

For the "DSA:LST+" architecture used in the alignment experiments, we add a cross-attention layer between the self-attention layer and the MLP layer of the side network. The cross-attention layer is followed by a layer-normalization operation before the output is fed into the MLP block. The cross attention uses the output of the self-attention layer of the adapter model to create the query vectors, and uses the hidden states of the base model to generate the key and value pairs. The hidden states of the base model are projected down to the size of the side model's hidden size using a low-rank linear projection with rank 8, before being fed into the cross attention block to generate the key-value pairs. To initialize this architecture, we pretrained it on the Dolma and Tulu-v3 datasets using the same training hyperparameters as we used for distilling the 'DSA:LST' adapter as described in the previous section. Compared to the distillation training run, the pretraining run differs in 2 ways: (1) the loss is the log-likelihood of the ground truth next-tokens without involvement of any teacher model (2) the logits output by the adapted model are created by taking the mean of the logits produced by the base model and the adapter.

In the "DSA:LLD" architecture, we try out low-rank adapters with rank 16, 32 and 64, and try out duplicating the last 1, 2 or 3 layers.

When evaluating different models on MMLU,MTBench, JailbreakBench and StrongReject benchmarks, we always use greedy decoding to generate the model responses.

When computing MMLU scores, which measure accuracy on the multiple choice questions, we use regular expressions to salvage and match model generated responses to ground truth as best as we can even if it does not match the asked format. This gives higher scores than using strict exact string matching for computing accuracy. This strategy is taken from the StrongReject repository[1].

---

[1] https://github.com/dsbowen/strong_reject/blob/main/src/mmlu_evaluation.py

## C   FLOP CALCULATION DETAILS

Table 5: Comparison of FLOPs overhead for different adaptation methods.

| Method | Base Model FLOPs | DSA FLOPs | Overhead Percentage |
|---|---|---|---|
| DSA:LLD (Zephyr) | 7,241,467,904,000 | 454,885,504,000 | 6.28% |
| DSA:LST+ (Zephyr) | 7,241,467,904,000 | 52,101,128,000 | 0.72% |
| DSA:LST (Zephyr) | 7,241,467,904,000 | 70,057,993,024 | 0.97% |
| DSA:PLR (Qwen-7B) | 7,170,637,888,000 | 7,168 | 0.00% |
| DSA:RTB (Qwen-7B) | 7,170,637,888,000 | 12,845,070,336 | 0.18% |
| DSA:LST (Qwen-7B) | 7,170,637,888,000 | 55,043,080,896 | 0.77% |

We compute the number of FLOPs for each model by running inference on 500 random tokens and using the in-built FlopCounterMode calculator in PyTorch Paszke et al. (2019). Detailed numbers of FLOPs for each model are provided in Table 5. Please note that for the LST-based architectures, the percentage overhead grows very slowly with the number of input tokens. For example, for 1k input tokens DSA:LST (Qwen-7B) adds 0.84% overhead increasing from 0.77% for 500 input tokens.

## D   FULL DATASET DETAILS

**ToxiGen Hartvigsen et al. (2022)**

**URL:** `https://huggingface.co/datasets/toxigen/toxigen-data`

**License:** MIT License

**Comments:** For the Toxigen dataset, we use the original train and test splits in the *annotated* subset of the dataset, except that we keep aside 5% of the train set to use as the validation set. The toxigen dataset has continuous labels given by humans on a scale of 1 to 5 denoting the toxicity. We filter the dataset to keep all examples with rating equal to 1 as the negative examples, and all examples with rating greater than or equal to 4 as positive examples. The final dataset contains 4418, 248 and 452 datapoints in train, validation and test splits.

**AEGIS2.0 Ghosh et al. (2025)**

**URL:** `https://huggingface.co/datasets/nvidia/Aegis-AI-Content-Safety-Dataset-2.0`

**License:** CC BY 4.0

**Comments:** For the Aegis dataset, we used the original train, validation and test splits containing 30k, 1.45k and 1.96k instances respectively of interactions along with safety labels. We use only the prompt and the label assinged to the prompt in our dataset. For prompts in the dataset are labeled by multiple annotators, we generate aggregated final label (safe or unsafe) for each prompt by taking majority vote. If equal number of ratings mark a prompt as safe and unsafe, we consider the final label to be unsafe.

**BeaverTails Ji et al. (2023)**

**URL:** `https://huggingface.co/datasets/PKU-Alignment/BeaverTails`

**License:** CC-BY-NC-4.0

**Comments:** For this dataset, we use the variant with 30k datapoints. We use the original test split containing 3k datapoints and set aside 10% of the train original split to use as validation set.

**Summedits Laban et al. (2023)**

**URL:** `https://huggingface.co/datasets/Salesforce/summedits`

**License:** CC-BY-4.0

**Comments:** We divide the dataset into train, validation and test splits ourselves consisting of 80%, 10% and 10% of datapoints of the entire dataset.

**Anthropic hh-rlhf Bai et al. (2022)**

**URL:** `https://github.com/anthropics/hh-rlhf`
**License:** MIT

**StrongREJECT Souly et al. (2024)**

**URL:** `https://huggingface.co/datasets/walledai/StrongREJECT`
**License:** MIT

**MTBench Zheng et al. (2023)**

**URL:** `https://github.com/lm-sys/FastChat/blob/main/fastchat/llm_judge/data/mt_bench/question.jsonl`
**License:** Apache-2.0

**Dolma Soldaini et al. (2024)**

**URL:** `https://huggingface.co/datasets/allenai/dolma`
**License:** ODC-BY
**Comments:** We use the officially released small subset of the entire Dolma corpus called *v1_6-sample* consisting of about 10 billion tokens.

**Tulu-v3 Lambert et al. (2024)**

**URL:** `https://huggingface.co/datasets/allenai/tulu-3-sft-mixture`
**License:** ODC-BY
**Comments:** We use 500K datapoints from the sft mixture for training.

**MMLU Hendrycks et al. (2021)**

**URL:** `https://huggingface.co/datasets/cais/mmlu`
**License:** MIT
**Comments:** We use the "all" subset and test split of the dataset containing about 14K datapoints for evaluation only.

**JailbreakBench Chao et al. (2024)**

**URL:** `https://huggingface.co/datasets/JailbreakBench/JBB-Behaviors`
**License:** MIT
**Comments:** We use the "behaviors" subset and "harmful" split of the dataset containing 100 datapoints for evaluation only.

# E  FULL MODEL ASSET DETAILS

**Qwen2.5-7B-Instruct (Yang et al., 2024)**

**URL:** `https://huggingface.co/Qwen/Qwen2.5-7B-Instruct`
**License:** Apache-2.0

**Qwen2.5-14B-Instruct (Yang et al., 2024)**

**URL:** `https://huggingface.co/Qwen/Qwen2.5-14B-Instruct`
**License:** Apache-2.0

**Mistral-7B-v0.3-Instruct (Jiang et al., 2023a)**

**URL:** `https://huggingface.co/mistralai/Mistral-7B-Instruct-v0.3`
**License:** Apache-2.0

**Gemma-2-9b-it (Team et al., 2024)**

**URL:** `https://huggingface.co/google/gemma-2-9b-it`
**License:** Gemma

**Zephyr-7B-beta (Tunstall et al., 2023)**

**URL:** `https://huggingface.co/HuggingFaceH4/zephyr-7b-beta`
**License:** MIT

## F DETAILS OF COMPUTE RESOURCES

Our experiments were run on a cluster of machines consisting of 8xH100 and 8xA100 GPUs from Nvidia, with each GPU equipped with 80GB of high-bandwidth memory. All experiments were run on single machines and no multi-node training was done for the experiments in this work. For training adapted models on classification tasks, the training time heavily depended on the dataset used and the number of epochs taken by the model to converge before the early stopping is executed, but typically finished within 2-3 hours for 7B models. The experiments for running DPO of adapted Zephyr models using the Anthropic helpful-harmless preference dataset took around 15-25 hours depending on the architecture and GPUs used. The distillation of 'DSA:LST' and Separate Guardrail models would take around 1.5-2 days on 8xA100 GPUs and pretraining the 'DSA:LST+' architecture for alignment experiments would take about 1 day on the same configuration.

