# OpenReview forum: "Disentangled Safety Adapters Enable Efficient Guardrails and Flexible Inference-Time Alignment"
_ICLR.cc/2026/Conference — Submitted to ICLR 2026_

### Official Review · Reviewer_Qsrg · 2025-10-20

**Soundness:** 3
**Presentation:** 3
**Contribution:** 3
**Rating:** 6
**Confidence:** 4

**Summary:**

This paper introduces Disentangled Safety Adapters (DSA), a modular approach to improving LLM safety without the trade-offs of guardrails or full alignment fine-tuning. DSA uses lightweight adapters that tap into the base model’s internal representations, enabling efficient and flexible safety control at inference time. Experiments on models like Qwen, Mistral, Gemma, and Zephyr show strong results.

**Strengths:**

1. The paper is clearly structured and highly readable. I think this paper effectively motivates the problem of AI safety trade-offs between guardrails and alignment fine-tuning, and communicates the key ideas in an intuitive way.

1. The concept of Disentangled Safety Adapters (DSA) offers a compelling alternative to existing paradigms; that is, separate guardrail models and alignment fine-tuning. By decoupling safety-specific computations while leveraging base model representations, DSA provides a modular and efficient framework for safety interventions at inference time. This represents a potentially important step forward in designing adaptive and resource-efficient safety systems for large language models.

1. The proposed method is technically solid and easy to follow. The authors provide a clear mathematical formulation for both DSA-based guardrails and alignment  and a thorough comparison to related architectures such as LoRA, Res-Tuning, and Ladder Side-Tuning. The description of targeted alignment effectively shows how context-dependent safety control can be achieved without retraining the base model.

1. The empirical results are impressive and demonstrate clear superiority over baseline methods across multiple datasets and model families.

**Weaknesses:**

1. The paper provides insufficient detail for reproduction. Although the appendices briefly list hyperparameters and dataset splits, the source code and trained models are not provided, and several important training settings (e.g., exact adapter dimensions, implementation details for cross-attention, initialization strategies, and data preprocessing) are only briefly summarized. Given the complexity of multi-module training (guardrail + alignment + targeted interpolation), reproducing the experiments from the text alone would be extremely difficult. The absence of public code limits the practical verification of the claimed efficiency and scalability benefits.

1. The paper indirectly evaluates helpfulness via MTBench scores, showing that DSA maintains most of the model’s helpfulness while improving safety. However, there is no direct human evaluation of response quality or real-world usefulness.

**Questions:**

1. Did the authors examine how the actual generated outputs change after applying DSA? If so, what qualitative characteristics or behavioral differences were observed in the model’s responses?

---

> ### Author Response · Authors · 2025-11-22
> **Official Comment by Authors**
>
> We thank the reviewer for their valuable feedback. We were happy to see that the reviewer found the paper “clearly structured and highly readable” and the proposed approach to be “technically solid and easy to follow”. We were also glad to see that the reviewer considered DSAs to be “a potentially important step forward in designing adaptive and resource-efficient safety systems” with empirical results that “demonstrate clear superiority over baseline methods”. Below, we respond to the questions asked and provide our comments on other issues raised in the review.
>
> ---
>
> ### Questions:
>
> > **Did the authors examine how the actual generated outputs change after applying DSA? If so, what qualitative characteristics or behavioral differences were observed in the model’s responses?**
>
>
> Thank you for this insightful question. In response to your inquiry, we have now conducted a qualitative examination of how DSA affects generated outputs, revealing distinct behavioral characteristics compared to baseline methods.
>
> We have uploaded a new Supplementary Material with qualitative analysis of sample outputs for 5 randomly sampled prompts from both StrongReject and MTBench, comparing base model, LoRA, and DSA:LLD:TA outputs.
>
> - *StrongReject analysis:* The base model (Zephyr-7B) complied with 4/5 harmful requests. Both LoRA and DSA:LLD:TA successfully avoid harm but via different mechanisms: LoRA generates explicit refusals, while DSA performs early stopping—initiating a response but terminating before producing harmful content.
>
> - *MTBench analysis:* DSA:LLD:TA maintains output quality comparable to the base model, as the alignment adapter is activated for only 5/160 prompts. Even when the alignment adapter is forced to be always on (λ=0)—a worst-case scenario—output quality remained strong, with premature truncation affecting only 1/5 cases.
>
> To investigate the role of the classifier DSA’s errors in the targeted alignment setting, we identified 5/160 MTBench queries falsely flagged as unsafe (provided in supplementary material, pages 13-14). We found that they typically involved sensitive topics (medical advice, domestic violence, bullying) discussed in safe contexts. This highlights the need for more nuanced guardrail classifier training data to reduce false alarms.
>
> These findings and samples will be added to the manuscript.
>
> ### Response to other issues raised:
>
> > **The paper indirectly evaluates helpfulness via MTBench scores, showing that DSA maintains most of the model’s helpfulness while improving safety. However, there is no direct human evaluation of response quality or real-world usefulness.**
>
> We acknowledge that human evaluation would be ideal. However, recruiting qualified annotators at scale for complex benchmarks like MTBench and StrongReject is very resource intensive. Both benchmark papers [1,2] validated their LLM-as-judge protocols against human raters, with MTBench reporting GPT-4 achieving inter-human agreement levels. Following standard practice in prior work including our primary baseline DeRA [3], we use LLM-as-judge evaluation.
>
> [1] Zheng, Lianmin, et al. "Judging llm-as-a-judge with mt-bench and chatbot arena." Neurips (2023)
>
> [2] Souly, Alexandra, et al. "A strongreject for empty jailbreaks." Neurips (2024)
>
> [3] Liu, Tianlin, et al. "Decoding-time realignment of language models." ICML (2024)

---

### Official Review · Reviewer_rFTf · 2025-10-31

**Soundness:** 2
**Presentation:** 2
**Contribution:** 2
**Rating:** 4
**Confidence:** 4

**Summary:**

The paper introduces Disentangled Safety Adapters (DSA), a novel framework that decouples safety-specific computations from the base model by attaching lightweight adapters that reuse internal representations. DSAs are applied to both safety guardrails and safety alignment. The authors show empirical results on various benchmarks, demonstrating that DSA guardrails outperform standalone classifiers with similar cost and that targeted alignment reduces the alignment tax compared to standard LoRA fine-tuning.

**Strengths:**

•⁠  ⁠Clear and Novel Concept: The DSA framework is an intuitive design that reuses the base model's computations for safety tasks, enabling low-cost, disentangled safety modules.
•⁠  ⁠Guardrail Performance: DSA guardrails match the performance of full-sized, high-cost models while adding minimal compute overhead (<1% FLOPs). They outperform comparable low-cost models, with up to a 53% relative AUC improvement on hallucination detection.
•⁠  ⁠Targeted Alignment Results: The method achieves safety levels comparable to standard LoRA fine-tuning while preserving ~98% of the base model's general task performance, effectively mitigating the alignment tax.
•⁠  ⁠Architectural Analysis: The paper provides clear justification for its design choices by comparing adapter architectures and iterating on alignment adapters.

**Weaknesses:**

•⁠  ⁠Novelty: While presented as a novel framework, DSA is largely an incremental adaptation of existing adapter and side-tuning ideas. The contribution lies mainly in applying these methods to safety tasks, which is valuable but not conceptually deep.
•⁠  ⁠Weak Baselines: Comparisons rely on low-cost guardrails rather than competitive guardrails (e.g., Llama-Guard 2). DSAs also receive richer initialization and distillation, compromising fairness. Stronger baselines and matched compute would be needed to validate the claimed efficiency advantage.
•⁠  ⁠Limited interpretability and diagnostic analysis: The paper offers no qualitative analyses, error breakdowns, or visualization of adapter activations. Without understanding what DSAs actually learn or when they fail, it is hard to assess their safety reliability or transparency.

**Questions:**

See weaknesses.

---

> ### Author Response · Authors · 2025-11-22
> **Official Comment by Authors**
>
> We thank the reviewer for their thoughtful feedback, which has substantially improved our work. We were happy to see that the reviewer considered the DSA framework to be a “Clear and Novel Concept”  and acknowledged that DSAs “outperform comparable low-cost models”, and can be seen “effectively mitigating the alignment tax”. We address the concerns individually below:
>
> ---
>
> > **Weak Baselines: Comparisons rely on low-cost guardrails rather than competitive guardrails (e.g., Llama-Guard 2).**
>
>
> We appreciate the suggestion to compare against state-of-the-art guardrails like LlamaGuard and commercial APIs. However, the primary contribution of our paper is to propose and validate a new architectural framework (DSA) that enables efficient and modular safety interventions, rather than to build a new SOTA guardrail model from scratch. To rigorously evaluate the benefits of our architecture, we designed controlled experiments where the only variable is the method used (DSA vs. a separate guardrail), while keeping the base model and training data constant. Unfortunately, direct comparisons with models like LlamaGuard or commercial APIs are not feasible for this purpose, as their training data and, in the case of APIs, model weights are not publicly available.
>
>
> > **DSAs also receive richer initialization and distillation, compromising fairness. Stronger baselines and matched compute would be needed to validate the claimed efficiency advantage**
>
> While DSA initialization via distillation is computationally heavy, this is a one-time cost for any base model, amortized across multiple downstream tasks. For instance, a single DSA:LST checkpoint was fine-tuned on 4 guardrail tasks (Table 2), spreading initialization costs across applications. Moreover, distillation is only required for LST-based adapters—other variants like DSA:LLD for alignment require no distillation-based initialization.
>
> > **⁠Limited interpretability and diagnostic analysis**
>
> We thank the reviewer for their suggestion on adding qualitative analysis of model outputs and their errors. In response, we have conducted a qualitative analysis examining specific examples of model behavior and failure cases.
>
> We have uploaded a new Supplementary Material with qualitative analysis of sample outputs for 5 randomly sampled prompts from both StrongReject and MTBench.
>
> - *StrongReject analysis*: The base model (Zephyr-7B) complied with 4/5 harmful requests. Both LoRA and DSA:LLD:TA successfully avoid harm but via different mechanisms: LoRA generates explicit refusals, while DSA performs early stopping—initiating a response but terminating before producing harmful content.
>
> - *MTBench analysis*: DSA:LLD:TA maintains output quality comparable to the base model, as the alignment adapter is activated for only 5/160 prompts. Even when the alignment adapter is forced to be always on (λ=0)—a worst-case scenario—output quality remained strong, with premature truncation affecting only 1/5 cases.
>
> - *Classifier false positives*: We identified 5/160 MTBench queries falsely flagged as unsafe, typically involving sensitive topics (medical advice, domestic violence, bullying) discussed in safe contexts (provided in supplementary material, pages 13-14). This highlights the need for more nuanced guardrail classifier training data to reduce false alarms.
>
> We will add these qualitative insights and samples to the paper.

---

> > ### Comment · Reviewer_rFTf · 2025-11-28
> >
> > Hi Authors,
> >
> > I appreciate your response. Please allow me until this weekend to read your responses carefully and provide feedback. Thank you!

---

### Official Review · Reviewer_yZaC · 2025-11-01

**Soundness:** 2
**Presentation:** 3
**Contribution:** 2
**Rating:** 4
**Confidence:** 4

**Summary:**

The paper focuses on two current technical approaches for LLM safety: guardrail models and safety alignment, with an emphasis on the trade-off between development flexibility and inference efficiency. It proposes a Disentangled Safety Adapter (DSA), which enhances safety by training independent guardrail and alignment adapter modules, without modifying the base model's parameters or altering its core inference path for general tasks.

**Strengths:**

1. The paper identifies the crucial trade-off between development flexibility and inference efficiency in current guardrail models and safety alignment methods, particularly in LLM deployment.
2. The DSA framework proposed in this paper unifies the training of guardrail models and safety alignment, which is an innovative approach.

**Weaknesses:**

1. Lack of Generalization Verification for the DSA Method:

 1.1. The experiments were conducted on only one base model, which itself has poor safety performance. It would be beneficial to see experiments on at least three different models, not just the base model, as models fine-tuned with instructions may be more prone to performance degradation after safety alignment.

1.2. The paper uses only one training dataset, but the introduction mentions one of the advantages of DSA as "updating safety policies without retraining or redeploying the base model." This claim requires validation with more diverse training data.

2. Missing Key Baselines:

The DSA framework proposed in this paper essentially adds additional modular parameters to the model, improving safety through optimization. There is a similar prior work, “MoGU: A Framework for Enhancing Safety of Open-Sourced LLMs While Preserving Their Usability,” but no comparison is made in the experiments.

3. Lack of Hyperparameter Analysis:

From Figure 2, it appears that the DSA method is quite sensitive to hyperparameters. Whether this impacts the method's practicality needs further analysis and clarification.

4. The necessary ablation analysis is lacking：

The proposed DSA Targeted Alignment framework includes two trainable modules: a guardrail module and an alignment module; however, there are no experiments directly comparing which of the two modules is more important.

**Questions:**

1. In the introduction, one of the advantages of DSA is mentioned as "updating safety policies without retraining or redeploying the base model," but the experiments lack additional validation for this point. I carefully reviewed the paper, and the authors only trained one model on one dataset.
2. Table 3 primarily compares DSA with LoRA-based DPO, but in fact, DSA increases the number of parameters compared to the base model, whereas LoRA-based methods do not add parameters during inference. This seems to contradict the motivation of the paper.
3. Additionally, LoRA's specific settings, such as hyperparameters like rank, can impact performance. It's unclear whether the authors considered this in their experiments. Why not consider full model fine-tuning?

---

> ### Author Response · Authors · 2025-11-22
> **Official Comment by Authors**
>
> We sincerely thank the reviewer for their constructive feedback and insightful questions. Below, we respond to the questions asked and provide our comments on other issues raised in the review.
>
> ---
>
> ### Questions:
>
> > **Table 3 primarily compares DSA with LoRA-based DPO, but in fact, DSA increases the number of parameters compared to the base model, whereas LoRA-based methods do not add parameters during inference. This seems to contradict the motivation of the paper.**
>
> Thank you for this important clarification opportunity. While DSA adds parameters, our primary motivation is reducing inference-time computational cost, not parameter count.
>
> LoRA-based flexible alignment (Liu et al., 2024) requires running the model twice, once with and once without LoRA, and then mixing outputs, totaling 200% compute. In contrast:
>
> * DSA:LST+:TA: 102% compute (base + harm classifier + aligned predictor)
> * DSA:LLD:TA: 108% compute
>
> Without flexible alignment, the LoRA baseline has no computational overhead but causes large drops in general performance to improve safety. Our targeted alignment methods achieve comparable or better safety with significantly less performance degradation and modest 2-8% additional compute overhead.
>
> We will clarify these compute requirements in the revision.
>
> ---
>
> > **Additionally, LoRA's specific settings, such as hyperparameters like rank, can impact performance. It's unclear whether the authors considered this in their experiments. Why not consider full model fine-tuning?**
>
> We systematically varied the rank for LoRA in increments of 8 using the Qwen2.5-7B-Instruct model and we found no benefit of increasing it beyond 32 (val set AUC scores in table below). Hence, we chose to run experiments with rank 32 for all models in the guardrail section.
>
> |LoRA rank|AEGIS2.0|BeaverTails|ToxiGen|Summedits|
> |---|---|---|---|---|
> 24|91.86|90.49|99.61|90.11
> 32|92.05|90.52|99.19|89.84
>
> Full fine-tuning is computationally very expensive and so we restricted ourselves to LoRA based training.
>
> ---
>
> ### Response to other issues raised:
>
> > **The experiments were conducted on only one base model, which itself has poor safety performance. It would be beneficial to see experiments on at least three different models, not just the base model, as models fine-tuned with instructions may be more prone to performance degradation after safety alignment.**
>
> We demonstrate DSA's effectiveness across multiple base models:
>
> Safety guardrails (Section 4.1): We evaluate across 4 different base models (Qwen2.5-7B, Qwen2.5-14B, Mistral-7B-v0.3, Gemma2-9B) with consistent improvements shown in Table 2.
>
> Safety alignment (Section 4.2): We deliberately chose Zephyr-7B because it lacks extensive safety fine-tuning. This is critical for isolating DSA's true impact—nearly all modern LLMs (including all models in Section 4.1) are already extensively safety-aligned by their developers. Using pre-aligned models would show smaller improvements from any alignment method, confounding our ability to measure efficacy. At the time of our experiments Zephyr-7B was the most capable publicly-available instruction-tuned model meeting this criterion.
>
> ---
>
> > **There is a similar prior work, “MoGU: A Framework for Enhancing Safety of Open-Sourced LLMs While Preserving Their Usability,” but no comparison is made in the experiments.**
>
> Thank you for highlighting this work! While MoGU also provides inference-time alignment with low compute overhead, it differs fundamentally from DSA:
>
> MoGU: Trains routers and adapters operating within base model layers, modifying key-value cache and hidden states "in-place" for all queries throughout generation (prefill and decode stages).
>
> DSA targeted alignment: Adapters operate on top of base model hidden states without modifying them. Base model cache and states remain unchanged. Adapter logits are used only when unsafe input is detected; otherwise discarded.
>
> Result: DSA perfectly preserves base model behavior on safe queries, maintaining clear factorization of safety from non-safety related behavior.
>
> We will add this discussion to related work in the final manuscript.
>
> ---
>
> > **The proposed DSA Targeted Alignment framework includes two trainable modules: a guardrail module and an alignment module; however, there are no experiments directly comparing which of the two modules is more important.**
>
> The guardrail and alignment modules serve complementary functions:
>
> - *Guardrail alone:* Detects harmful queries but cannot generate safe responses.
>
> - *Alignment alone:* Produces aligned logits that can be mixed with base model logits (weight λ) for safe responses. However, applying this to all queries causes significant capability drops (see MTBench scores in Figure 2 for non-TA methods).
>
> - *Combined (TA):* Using the guardrail to selectively apply the alignment module substantially reduces capability degradation. Figure 2 demonstrates this difference across λ values (DSA:LST+ vs DSA:LST+:TA and DSA:LLD vs DSA:LLD:TA).

---

### Official Review · Reviewer_7YFw · 2025-11-01

**Soundness:** 2
**Presentation:** 2
**Contribution:** 2
**Rating:** 4
**Confidence:** 3

**Summary:**

This paper introduces Disentangled Safety Adapters (DSA), a framework that decouples safety-specific computations from task-optimized base models using lightweight adapters. The authors propose DSA for two applications: (1) safety guardrails that classify unsafe inputs/outputs by reusing base model representations, and (2) safety alignment that steers generation by interpolating between base model and adapter logits. The key innovation is "targeted alignment" that dynamically adjusts alignment strength based on input safety classification. Experiments show DSA guardrails outperform comparably-sized standalone models (up to 53% AUC improvement on hallucination detection) while adding <1% computational overhead, and DSA targeted alignment reduces alignment tax by 8 percentage points compared to standard fine-tuning.

**Strengths:**

1. Well-motivated problem: The paper clearly articulates the trade-off between separate guardrails (flexible but expensive) and alignment training (efficient but inflexible), positioning DSA as an elegant solution combining advantages of both approaches.
2. Good experimental evaluation:
    - Multiple base models (Qwen, Mistral, Gemma) and scales (7B-14B)
    - Diverse safety tasks (hate speech, unsafe prompts/responses, hallucination detection)
    - Systematic architecture comparison (PLR → RTB → LST)
    - Thorough ablations and multiple benchmarks
3. Good empirical results: DSA:LST matches full-sized guardrails at <1% overhead. Consistent improvements across model families and tasks

**Weaknesses:**

1. Limited architectural novelty: The paper primarily applies existing adapter architectures (Side-Tuning, LST, Res-Tuning) to safety tasks. While the application is novel, the core technical contribution is incremental. The DSA:LST+ and DSA:LLD variants introduced for alignment are relatively minor modifications.

2. Binary switching mechanism: Using a step function ($\lambda \in \{0,1\}$) is simplistic. The paper mentions smooth interpolation as future work, but this seems essential for robustness.

3. DSA:LST and DSA:LST+ significantly underperform LoRA baseline (Figure 2). And I don't agree the statement that
```
This method (Fig. 2, "LORA") incurs no additional inference cost but entangles safety with
base model computations, precluding flexible alignment strength adjustment without running both
aligned and unaligned models
```
Actually, LoRA can be only used in the last layer and most computaions can be used.

4. Limited baseline comparisons: No comparison with other efficient guardrail methods. Is there any other methods for similar tasks? For example, a good and simple baseline could be just prompt engineering: Prompting LLM itself to output whether the input/output is safe or not, and most computations could be reused due to prompt KV caching mechanism.

**Questions:**

1. What is the actual latency overhead in your experiments and in production settings with batching?
2. Can adapters trained for one safety policy be efficiently adapted to new policies?

---

> ### Author Response · Authors · 2025-11-22
> **Official Comment by Authors**
>
> We thank the reviewer for their constructive feedback and insightful questions. We were happy to see they liked the paper’s “well-motivated problem”, “good experimental evaluation” and “good empirical results”. We are particularly grateful for the reviewer's encouragement to add the prompting baseline, which strengthens our work by demonstrating the advantage of learned adapters over zero-shot prompting approaches. Below, we respond to the questions asked and provide our comments on other issues raised in the review.
>
> ---
>
> ### Questions:
>
> > **What is the actual latency overhead in your experiments and in production settings with batching?**
>
> We measure the wall time (in seconds) for single forward pass on batch size 32, sequence length 1000 using randomly sampled token inputs. We run everything in bfloat16 precision. The results are in the table below:
>
> |Model|CPU-time|CUDA-time|
> |---|---|---|
> |Base Model(Qwen-7B)|1.966|1.960|
> |DSA:LST (Qwen-7B)|2.072|2.065|
> |Base Model (Zephyr)|2.197|2.187|
> |DSA:LLD (Zephyr)|2.350|2.345|
>
> There is about 5% latency overhead for running the LST adapter, and about 7% overhead for the LLD adapter. This is substantially less than the 100% overhead of running a separate guardrail of comparable performance.
> Additionally, our current implementation is designed to test the effectiveness of the DSA architectures on downstream tasks (e.g. harm classification), it is not optimized for fast inference and could likely be parallelized more.
>
> ---
>
> > **Can adapters trained for one safety policy be efficiently adapted to new policies?**
>
> Since the DSA adapter training is cheap and disentangled from base model training, efficient adaptation to a new safety policy is supported by design.
>
> ---
>
> ### Response to other issues raised:
>
> > **Binary switching mechanism: Using a step function ( $\lambda \in \{0,1\}$) is simplistic. The paper mentions smooth interpolation as future work, but this seems essential for robustness.**
>
>
> The binary switching is a special case of the general framework (Eq. 3) where λ = f(classifier_output). We establish that even a simple step function works well as shown in Table 3. For example, the DSA:LST+:TA adapter reduced Strongreject score from 0.58 to 0.04 while the MTBench score decreased from 7.07 to 6.89. This already reflects better robustness than LoRA which got the same Strongreject score but suffered about 4x greater drop in MTBench score. The advantage of a step function is that base model behavior is guaranteed to be preserved on all non-safety related inputs. A smooth switching function would sacrifice this strong factorization between safety and non-safety specific behavior.
>
> ---
>
> > **Actually, LoRA can be only used in the last layer and most computaions can be used.**
>
> We agree, and that forms the basis of the DSA:LLD architecture that we propose in the work. If a LoRA is only used in the last k layers out of total N layers in the transformer, we can reuse the computation of the first N-k layers and run the last k layers twice (one with LoRA activated and one with it deactivated). This is exactly what DSA:LLD does. To the best of our knowledge, this approach has not been used for efficient and controllable safety alignment and is a contribution of our work.
>
> ---
>
> > **a good and simple baseline could be just prompt engineering: Prompting LLM itself to output whether the input/output is safe or not, and most computations could be reused due to prompt KV caching mechanism.**
>
> We thank the reviewer for suggesting this baseline. We implemented it by appending the guardrail prompt after the user query (for Aegis and Toxigen datasets) and after model response (for Summedits and Beavertails datasets).
>
> |Base model|Method|ToxiGen|AEGIS2.0|BeaverTails|Summedits|
> |---|---|---|---|---|---|
> |Qwen2.5-7B|Prompting|0.88|0.76|0.57|0.64|
> |Qwen2.5-7B|DSA:LST|0.98|0.93|0.93|0.88|
> |Qwen2.5-14B|Prompting|0.85|0.82|0.73|0.71|
> |Qwen2.5-14B|DSA:LST|0.99|0.94|0.92|0.88|
> |Mistral-7B-v0.3|Prompting|0.95|0.79|0.79|0.62|
> |Mistral-7B-v0.3|DSA:LST|0.98|0.93|0.93|0.90|
> |Gemma2-9B|Prompting|0.98|0.87|0.86|0.78|
> |Gemma2-9B|DSA:LST|0.98|0.93|0.93|0.89|
>
> We see that it always underperforms substantially compared to DSA:LST adapters except on Toxigen with the Gemma2-9B base model.
> Note that we cannot improve the prompting baseline through training because that would require fine-tuning the base model on safety labels, directly violating the premise of our work: maintaining an unmodified base model which can produce exactly same responses to queries. The prompting approach is inherently limited to the base model's zero-shot reasoning capacity, whereas DSA:LST learns task-specific safety features through training while preserving the base model's general capabilities.
> This comparison actually strengthens the case for DSA: it shows that simply prompting an LLM is insufficient for robust safety classification, and that learned adapters perform much better.

---

### Meta-Review · Area_Chair_gPHj · 2026-01-07

**Summary:**

The paper proposes Disentangled Safety Adapters (DSA), a modular framework that attaches lightweight adapters to a frozen base model to support efficient safety guardrails and inference-time targeted alignment. The work is well motivated, clearly written, and supported by extensive experiments across several model families and safety tasks. However, the technical novelty is incremental, relying largely on known adapter or similar ideas, and the evaluation lacks stronger competitive baselines and clearer reproducibility details. I thus recommen rejection for this paper.

**Reviewer Concerns:**

Most major concerns raised by reviewers 7YFw and Qsrg regarding missing prompting baselines, qualitative analysis, and behavioral differences were directly addressed in the rebuttal with new experiments and supplementary analysis.

However, concerns about limited novelty, missing comparisons to stronger guardrails, lack of generalization validation across datasets and policies remain only partially addressed or largely outstanding.

**Reviewer Scores:**

Reviewer 7YFw and Qsrg would likely maintain or slightly increase the score (with the newly added experiments). Reviewers yZaC and rFTf would likely keep their original borderline scores, as their key concerns about novelty is still outstanding.

---

### Decision · Program_Chairs · 2026-01-26

Reject